# Predicting Heart Rate Slow Component Dynamics: A Model Across Exercise Intensities, Age, and Sex [note 1]

**DOI:** 10.3390/sports13020045

**Published:** 2025-02-07

**Authors:** Massimo Teso, Alessandro L. Colosio, Maura Loi, Jan Boone, Silvia Pogliaghi

**Affiliations:** 1College of Health and Life Sciences, Hamad Bin Khalifa University, Doha P.O. Box 34110, Qatar; 2Laboratoire Interuniversitaire de Biologie de la Motricité, Jean Monnet University, 42100 Saint-Étienne, France; alessandro.colosio@univ-st-etienne.fr; 3Department of Neurosciences, Biomedicine and Movement Sciences, University of Verona, 37129 Verona, Italy; maura.loi@univr.it; 4Department of Movement and Sports Sciences, Ghent University, Watersportlaan 2, 9000 Ghent, Belgium; jan.boone@ugent.be

**Keywords:** heart rate, exercise test, aerobic exercise, slow component, HR drift, cardiovascular drift

## Abstract

The heart rate slow component (_sc_HR) is an intensity-dependent HR increment that emerges during constant exercises, partially dissociated from metabolism ($\dot{V}$O_2_). The _sc_HR has been observed during constant-workload exercise in young and older adults. Unless this _sc_HR is accounted for, exercise prescription using HR targets lead to an undesired reduction in metabolic intensity over time. Purpose: The purpose of this study is to characterize _sc_HR across intensities, sex, and age to develop and validate a predictive equation able to maintain the desired metabolic stimulus over time in a constant aerobic exercise session. Methods: In our study, 66 individuals (35 females; 35 ± 13 yrs) performed the following: (i) a ramp-test for respiratory exercise threshold (GET and RCP) and maximal oxygen uptake ($\dot{V}$O_2max_) detection, and (ii) 6 × 9-minute constant exercises at different intensities. The _sc_HR was calculated by linear fitting from the fifth minute of exercise (bpm⋅min^−1^). A multiple-linear equation was developed to predict the _sc_HR based on individual and exercise variables. The validity of the equation was tested on an independent sample by a Pearson correlation and Bland–Altman analysis between the measured and estimated HR during constant exercises. Results: The _sc_HR increases with intensity and is larger in males (*p* < 0.05). A multiple-linear equation predicts the _sc_HR based on the relative exercise intensity to RCP, age, and sex (*r*^2^ = 0.54, SEE = 0.61 bpm⋅min^−1^). _sc_HR (bpm⋅min^−1^) = −0.0514 + (0.0240 × relative exercise intensity to RCP) − (0.0172 × age) − (0.347 × Sex (males = 0 and females score = 1)). In the independent sample, we found an excellent correlation between the measured and estimated HR (r^2^ = 0.98, *p* < 0.001) with no bias (−0.01 b·min^−1^, z-score= −0.04) and a fair precision (±4.09 b·min^−1^). Conclusions: The dynamic of the _sc_HR can be predicted in a heterogeneous sample accounting for the combined effects of relative intensity, sex, and age. The above equation provides the means to dynamically adapt HR targets over time, avoiding an undesired reduction in the absolute and relative training load. This strategy would allow the maintenance of the desired metabolic stimulus ($\dot{V}$O_2_) throughout an exercise session in a heterogeneous population.

## 1. Introduction

Aerobic physical activity induces specific metabolic stimuli and adaptations able to improve the cardiometabolic fitness in a given individual in a dose–response manner [1,2]. As such, aerobic exercise is a fundamental ingredient of any training intervention for health promotion [3]. The exercise prescription dose is typically quantified by these four elements: frequency, intensity, time, and type of exercise (according to the FITT scheme) [4]. Frequency, time, and type are relatively easy to determine, manipulate, and monitor, while intensity remains the most complex and elusive term of an exercise prescription dose [5,6].

Exercise intensity can be expressed in “absolute” or “relative” terms [7]. Absolute intensity refers to the energy required to perform a specific activity, and, for aerobic exercise, this can be measured through a metabolic equivalent as the oxygen uptake [5]. On the contrary, relative intensity refers to the stress imposed on the body’s homeostasis during exercise and is typically expressed as a percentage of anchor measurements, such as the maximal or reserve oxygen uptake (%$\dot{V}$O_2max_ and %$\dot{V}$O_2R_) [5,8]. The implementation of either the absolute or relative exercise intensity outside of a laboratory environment typically entails the transition of the metabolic equivalent into an external load such as speed, watt, or pace that elicits the desired metabolic intensity (e.g., $\dot{V}$O_2_ or %$\dot{V}$O_2max_) [9]. Whenever this approach is impossible or impractical, the heart rate (HR) is commonly used as an easy-to-measure and inexpensive intensity index to prescribe and monitor the exercise intensity in both clinical and sports settings [10]. The prescription of exercise intensity using HR targets relies on the existence and constancy over time of a linear relationship between the HR and oxygen uptake (as either $\dot{V}$O_2_ or %$\dot{V}$O_2max_) [11,12]. Accordingly, guidelines for exercise intensity prescription typically include HR targets (%HR_max_ or %HRreserve) intended to generate specific metabolic stimuli and, in turn, specific training adaptations [3].

However, recent studies have raised our awareness of a problem that has been underappreciated: during prolonged constant-work exercise, a time-dependent mismatch emerges between HR and oxygen consumption as a result of a slow rise in HR, partially independent of metabolism (i.e., heart rate slow component) [13,14,15,16]. This phenomenon is present from the moderate to the severe domains, with an amplitude that appears larger with increasing intensity [13,14,15]. On the one hand, the failure of the heart rate to attain a steady state response hinders the accurate association of a univocal %HR_max_ or HR target to any exercise intensity [6,13]. On the other hand, when a constant HR target is maintained over time, the workload will be progressively reduced throughout the exercise session; this, in turn, will lead to an undesired reduction in the metabolic stimulus that was intended to be constant [6,15,16,17]. Therefore, the practical impact of ignoring the HR slow component when prescribing exercise (anchored to an %HR_max_ or HR target) may be minimal or severe depending on its amplitude, which can differ across intensities and in different populations, fitness levels, or diseases [13,14,16,17].

The heart rate slow component (_sc_HR) has been repeatedly described in male populations (i.e., healthy adults [14,17], and adolescents suffering from obesity [15]), and in a group of postmenopausal women [13]. Interestingly, in postmenopausal women, the _sc_HR was found to be one-third smaller compared to what was reported for adult males [13,14]. In our former study [13], we hypothesized that the observed smaller dynamic of HR could be related to (i) a higher potential for HR excursion (i.e., a larger HR reserve) in younger individuals compared to older ones, and/or (ii) a higher absolute heat production in younger and fitter males compared to older females that may have affected the core temperature over time.

Regarding the possible aging effect, a study conducted on obese adolescents (15) found an _sc_HR slightly lower compared to what was reported for adult males [14] (respectively, in the moderate and heavy domain: ≈0.4 and ≈2.4 vs. ≈0.9 and ≈2.9 bpm⋅min^−1^). Nevertheless, this discrepancy might not have been derived from the age differences but from the lower catecholamine response affecting individuals suffering from obesity that, in turn, might have blunted the HR dynamics [15].

In addition, 3 weeks of moderate aerobic training has been shown to attenuate the _sc_HR [15]; the findings appear to suggest an effect of the cardiorespiratory fitness level on the _sc_HR. As such, the aging $\dot{V}$O_2max_ decline (about ~10% every 10 years [18]) might, therefore, reduce the _sc_HR in older individuals compared to young ones. However, nowadays, no studies have tested these hypotheses.

Lastly, in our former study, we also proposed that HR targets could be adjusted over time to ensure a constant metabolic intensity; with this aim, we provided a population-specific equation to predict the _sc_HR in postmenopausal women [13]. However, a model with external validity needs to be developed in order to grant an accurate prediction of _sc_HR at a given intensity, accounting for the possible role of sex, age, and fitness level in this dynamic.

To this aim, we performed a two-step study: (i) the development of a comprehensive model for _sc_HR prediction across different intensities, sexes, ages, and cardiorespiratory fitness, and (ii) testing the validity of the developed model using an independent sample of individuals. We hypothesized that the intensity, sex, age, and cardiorespiratory fitness level would predictably affect the HR slow component, allowing for the HR estimation over time in the independent sample. If confirmed, the developed mathematical model would allow for adjusting the prescribed HR targets over time by accounting for the individually estimated slow component of HR and, in turn, maintaining the desired metabolic stimulus during prolonged sessions across individuals.

## 2. Materials and Methods

### 2.1. Participants

A total of one hundred and one healthy subjects were recruited by advertisement within the local community and agreed to participate in this two-step study.

The size required for each step was determined based on the power analysis reported above in the statistics analysis section. In addition, females and males were equally subdivided into three age groups (young <36, middle-aged between >36 and <55, and elderly >55 years [19]). As a result, sixty-five individuals (23 young, 12 females; 22 middle-aged, 12 females; and 20 elderly, 10 females) participated in *Step 1, development of the prediction equation,* while thirty-six (12 young, 6 females; 12 middle age, 6 females; and 12 elderly 6 females) participated in *Step 2, validation of the prediction equation* (see the respective results sections forparticipants’ characteristics). Inclusion criteria were individuals of both sexes and age > 18 years. Exclusion criteria were smoking and any medical condition or therapy that could influence the physiological responses during testing. Moreover, they were fully informed about the study procedures and the potential risks and discomfort associated with the exercise testing before agreeing to sign a written informed consent form. The study was approved by the Ethics Committee of the University of Verona (CARP) and conducted in conformity with the Declaration of Helsinki (no. 16-2019).

### 2.2. Study Design

After medical clearance, the subjects’ main anthropometric measurements were collected (body mass (digital scale, Seca877, Seca, Leicester, UK) and height (vertical stadiometer, Seca, Leicester, UK)) [20].

During Visit 1, all participants performed a ramp incremental test until volitional exhaustion on an electromagnetically braked cycle ergometer (Sport Excalibur, Lode, Groningen, The Netherlands) for the determination of gas exchange threshold (GET), respiratory compensation point (RCP), and maximal parameters ($\dot{V}$O_2peak_ and HR_peak_).

On the successive appointments, subjects performed the following constant work rate exercises: *Step 1*, during which participants performed six 9-minute constant-work-rate exercises: two below GET (i.e., moderate domain), two between GET and RCP (i.e., heavy domain), and two above RCP (i.e., severe domain); and *Step 2*, during which participants performed three constant-work-rate exercises lasting 15 min or until exhaustion: one in each domain (i.e., from the moderate to the severe domain). The constant-work-rate order was randomized and counterbalanced. Participants were instructed to avoid caffeine consumption and physical activity at least 8 h and 24 h before each visit, respectively [21]. All visits were separated by at least 48 h and completed within 30 days. Tests were conducted at the same time of the day (±2 h) in an environmentally controlled laboratory (22–25 °C, 55–65% relative humidity). The cycloergometer position was set at the first visit to the lab and recorded for successive tests. To minimize the variability of glycogen oxidation, participants consumed a standardized meal (i.e., 500 cc of water and 2 g⋅kg^−1^ of low-glycaemic-index carbohydrates) two hours before attending the laboratory [22].

### 2.3. Ramp Incremental Protocol

The ramp incremental test consisted of 6 min baseline cycling at 50–100, followed by 4 min baseline cycling at 20–40 W to calculate the mean response time (MRT) (see data analysis); thereafter, power output increases by 10–30 W every minute until volitional exhaustion [23]. The warm-up load and ramp increment were customized depending on the estimated fitness level of each subject to reach the individual’s exhaustion between 8 and 12 min, as described in detail elsewhere [24]. Participants were asked to pick a self-selected cadence between 70–90 rpm and to maintain it for all tests. Breath-by-breath pulmonary gas exchange and HR were continuously measured using a metabolic cart (Quark B2, Cosmed, Rome, Italy). Moreover, for Step 1 only, the rating of perceived effort was collected using a Borg 6–20 scale 20 min from the end of the ramp incremental test [25].

### 2.4. Constant-Work Protocol

All the constant-work-rate exercises were preceded by a 3 min freewheeling cycling warm-up followed by an instantaneous increase in power output.

For *Step 1*, the exercise intensity of the six trials was chosen as follows:(i)Moderate trials: 33% (M1) and 66% (M2) of the difference between rest $\dot{V}$O_2_ and $\dot{V}$O_2_ at GET;(ii)Heavy trials: 33% (H1) and 66% (H2) of the difference between $\dot{V}$O_2_ at GET and RCP;(iii)Severe trials: 33% (S1) and 66% (S2) of the difference between $\dot{V}$O_2_ at RCP and $\dot{V}$O_2max_.


For *Step 2*, the exercise intensity was chosen as follows:(i)Moderate trials: 50% of the difference between rest $\dot{V}$O_2_ and $\dot{V}$O_2_ at GET;(ii)Heavy trials: 50% of the difference between $\dot{V}$O_2_ at GET and RCP;(iii)Severe trials: 50% of the difference between $\dot{V}$O_2_ at RCP and $\dot{V}$O_2max_.

To identify the power output that elicits the above $\dot{V}$O_2_ targets, the individual $\dot{V}$O_2_/power output relationship derived from the incremental exercise was corrected for the $\dot{V}$O_2_ mean response time and slow component by applying the mathematical model recently proposed by Caen et al. [9].

Breath-by-breath pulmonary gas exchange, ventilation, and HR were continuously measured using the same method of the ramp incremental protocol. Moreover, the perceived effort ratings (RPEs) were collected using a Borg 6–20 scale at the fifth minute during each constant-work-rate exercise [25].

### 2.5. Data Analysis

Breath-by-breath gas exchange variables and HR were treated as follows: aberrant data points (that lay 3 standard deviations away from the local mean) were removed [26]. This was accomplished using a linear least-squares regression method whereby the baseline (fitting window: approximately −180 to 0 s) and steady-state period (fitting window: approximately 180– end trial) were fitted. The 99% prediction bands were used to identify any data points that lay three SD from the local mean. Care was taken not to delete data in the early portion of the transition (i.e., <180 s); thereafter, data were linearly interpolated at 1 s, and then mediated at 5 s intervals [27]. GET and RCP were determined from the ramp incremental test by three experts independently using the standard technique [28]. $\dot{V}$O_2max_ (absolute and relative to the body weight), the maximal respiratory exchange ratio (RER_max_), and HR_max_ were determined as, respectively, the average $\dot{V}$O_2_ and RER of the last 30 s and the highest HR achieved before exhaustion during the ramp incremental protocol. The steady-state $\dot{V}$O_2_, measured during the 50–100W bouts prior to the ramp incremental test, was used to correct the $\dot{V}$O_2_/power output relationship for the mean response time [29]. To correct the ramp-identified power output above the gas exchange threshold, an additional correction to account for the $\dot{V}$O_2_ slow component was applied [9]. The power outputs associated with the target $\dot{V}$O_2_ were obtained using the mathematical model developed by Caen et al. [9].

In each constant work rate, we calculated the following:

For *Step 1*: (i) oxygen pulse ($\dot{V}$O_2_/HR) at the fifth minute and ninth minute of exercise; and (ii) _sc_HR, as the slope of the HR/time linear fitting from the fifth minute to the end of the exercise and expressed in both absolute units (bpm⋅min^−1^) and relative to the $\dot{V}$O_2_ at the fifth minute (_sc_HR/_5min_$\dot{V}$O_2_);

For *Step 2*: the mean HR value at the fifth minute and last minute of exercise.

### 2.6. Statistical Analysis

All data are presented as mean ± SD. After assumption verification (i.e., normality, and homogeneity of variance), the within-subject coefficient of variation and a two-way repeated-measure ANOVA (trial × intensity) were used to evaluate HR data repeatability measured during each freewheeling.

A mixed RM-ANOVA was performed to compare the values at the fifth minute of power output, $\dot{V}$O_2,_ HR, and %RCP, as well as the value of RPE, across exercise intensities, with sex as between-subjects factor.

To verify the presence and the amplitude of _sc_HR and its relationship with $\dot{V}$O_2_, $\dot{V}$O_2_/HR between the 5th and the 9th minute was compared across intensities by a two-way RM-ANOVA (time × intensity). Moreover, _sc_HR and _sc_HR/_5min_$\dot{V}$O_2_ were compared by a mixed RM-ANOVA across intensities with sex as a between-subjects factor. A post hoc analysis was performed using the Holm–Sidak method.

To develop a multi-linear model for the prediction of the individual _sc_HR, we proceeded as follows: (i) the sex parameter was classified as male = score 0 and female = score 1; (ii) relative exercise intensity to RCP (%RCP) was calculated based on the mean individual $\dot{V}$O_2_ measured at the 5th minute for each trial; and (iii) a forward multiple-linear regression was initially run including these variables: age, sex, %RCP, HR_max_, absolute $\dot{V}$O_2_ at the 5th minute, and relative $\dot{V}$O_2max_ to body weight. This analysis identified non-significant (*p* > 0.05) and cross-correlated predictors (i.e., correlation coefficient > 0.70, and variance inflation factor > 5) that were discarded from the model. Subsequently, a forward multiple regression was rerun until significant, non-cross-correlated predictors were identified, and the best prediction model was found.

To test the validity of the developed equation in *Step 2*, we proceed as follows: (i) verify the presence of an _sc_HR by comparing the HR measure at the 5th minute with the last minute of exercise by a paired t-test; (ii) estimate the HR at the end of the exercise with the following formula:HR@end = HR@5min + (_sc_HR⋅(time − 5))
where _sc_HR was estimated using the previously developed equation derived from *Step 1,* and time is in minutes from the start of the exercise; and, (iii) lastly, compare the measured and the estimated HR at the end of all exercises by a mixed RM-ANOVA (method × sex), Pearson correlation, and Bland–Altman analysis.

A power analysis was conducted a priori (G*Power 3.1). To develop a valid prediction model, the sample size required for *Step 1* was 60 individuals. Moreover, based on the standard deviations of the primary outcomes (_sc_HR for *Step 1* and within-subject variability of HR for *Step 2*) detected in previous studies [13,14,15], a minimum of 18 subjects were required to identify significant differences with an α error of 0.05 and a statistical power (1 − β) of 0.80. All statistical analyses were performed using SigmaPlot (version 14.0). Statistical significance was accepted when *p* < 0.05.

## 3. Results

*Step 1.* Participants’ characteristics and maximal parameters derived from the ramp incremental test are reported in Table 1. The average body mass index and cardiorespiratory variables were indicative of a normal weight and active population [4]. The two-way ANOVA on the HR during freewheeling shows no significant effect among the six trials (*p* = 0.11). The mean within-subject coefficient of variation was 2.7 ± 2.6%.

Participants’ HR and $\dot{V}$O_2_ responses during the constant-workload exercises are shown in Figure 1, while variables at the fifth minute are reported in Table 2. A mixed RM-ANOVA on power output, $\dot{V}$O_2,_ HR, %RCP, and RPE found, as expected, a significant effect of the relative exercise intensity (for all variables *p* < 0.001), while a significant main effect regarding sex was found for the power output and $\dot{V}$O_2_ only (power output and $\dot{V}$O_2_: *p* < 0.01; HR: *p* = 0.11; %RCP: *p* = 0.39; RPE: *p* = 0.28).

A two-way RM-ANOVA on $\dot{V}$O_2_/HR found a significant main effect for both time and intensity (respectively, *p* < 0.05 and *p* < 0.01). A post hoc analysis for time shows a lower $\dot{V}$O_2_/HR at the ninth minute compared to the fifth minute in all the intensities from the moderate to the heavy domain (mean difference ±SD: M1 −0.17 ± 0.63, M2 −0.32 ±0.68, H1 −0.45 ±0.60, and H2 −0.60 ± 0.57 mL⋅b^−1^). In contrast, no difference was found in both intensities in the severe domain (mean difference: S1 −0.13 ± 0.74, and S2 0.14 ± 0.72 mL⋅b^−1^).

The _sc_HR in absolute values and relative to the fifth minute of $\dot{V}$O_2_ are shown in Figure 2. A mixed RM-ANOVA on _sc_HR found a significant main effect of relative exercise intensity and sex (respectively, *p* < 0.001 and *p* < 0.05) and no interaction (*p* = 0.48). Moreover, when the _sc_HR was expressed relative to the _5min_$\dot{V}$O_2,_ a significant main effect of intensity was confirmed (*p* < 0.001), while the effect of sex disappeared (*p* = 0.24) with no interaction (*p* = 0.18). The post hoc analysis for sex within each intensity is shown in Figure 2. The post hoc analysis for intensity within each domain (i.e., S2 vs. S1, H2 vs. H1, and M2 vs. M1) showed no difference for either _sc_HR or _sc_HR/_5min_$\dot{V}$O_2_ (*p* > 0.05 for all comparisons).

The iterative application of the forward multiple-linear regression to estimate the _sc_HR indicated that all the analyzed factors were significant predictors of the _sc_HR (*p* < 0.05). However, the $\dot{V}$O_2_ at the fifth minute, HR_max_, and relative $\dot{V}$O_2max_ to body weight were cross-correlated with %RCP and/or age (r^2^ > 0.70 and variance inflation factor >10) and discarded from the analysis as their removals did not significantly reduce the explanatory power of the model (i.e., *p* and r^2^). Thus, the most relevant and not cross-correlated predictor of the _sc_HR was %RCP along with sex and age (*p* < 0.01). Then, the following predicting equation for the _sc_HR was found:_sc_HR (bpm⋅min^−1^) = −0.0514 + (0.0240 ⋅ intensity expressed in %RCP) − (0.0172 ⋅ age) − (0.347 ⋅ Sex (males = score 0 and females = score 1))
r^2^ = 0.53; standard error of estimate = 0.61

*Step 2.* To investigate the external validity of the present equations, we tested the model on an independent sample (Table 3 for individual features).

A paired t-test showed significant increments in the HR from the fifth minute to the end (12 ± 3 min) of the exercise (HR mean of the three intensities at, respectively, the fifth minute, 129 ± 21 b⋅min^−1^, and the end of the exercise, 136 ± 24 b⋅min^−1^, *p* < 0.05).

A mixed RM-ANOVA showed higher end-exercise HR in females compared to males (*p* < 0.001); however, it showed no difference between the measured and estimated HR (*p* = 0.86) and no interaction (sex × method, *p* = 0.82) (respectively, the mean of the three intensities for the estimated and directly measured HR for females is 146 ± 27 and 145 ± 26 b⋅min^−1^; and males, 126 ± 19 and 127 ± 18 b⋅min^−1^). Figure 3 displays the correlation and Bland–Altman analysis performed between the measured and estimated end-exercises’ HR, indicating an excellent correlation and correspondence with the small, non-significant bias of −0.01 ± 4.09 b⋅min^−1^ (LoA: lower = −7.84 b⋅min^−1^ and upper = 7.81 b⋅min^−1^; z-score = −0.04) that was within the day-to-day variability detected in the present study for HR values: 2.7 ± 2.6%.

## 4. Discussion

The purpose of this study was to develop a comprehensive prediction model for the _sc_HR across exercise intensities in both sexes and different ages and to test the validity of the developed model using an independent sample of individuals.

The study confirmed the presence of an _sc_HR in moderate, heavy, and severe domains, which proportionally increases with relative intensity. Moreover, the study demonstrated, for the first time, a significant effect of sex and age on the amplitude of the _sc_HR. Finally, the study developed a comprehensive equation that, by including %RCP, sex, and age, accurately predicted the _sc_HR in an independent sample of males and females of different ages across the three domains.

The individual and anthropometric characteristics of the subjects enrolled in the study were in line with what we expected from the existing literature for healthy, active individuals [3].

In agreement with previous findings on either male-only [14,15] or female-only [13] populations, our data confirmed the presence of the _sc_HR from the moderate to the severe domain and confirmed its relationship with the relative exercise intensity [13,14,15,16].

However, in male individuals, previous studies described a higher _sc_HR compared to our male sample (i.e., mean, respectively, of ≈0.9, ≈2.9, and ≈6.7 bpm⋅min^−1^ versus ≈0.55, ≈1.35, and ≈2.2 bpm⋅min^−1^, for the moderate, heavy, and severe domain) [14]. On the contrary, in females, the _sc_HR in the present study was approximately double the size of the values reported previously in postmenopausal women (i.e., mean, respectively, of ≈0.22, ≈0.99, and ≈1.8 bpm⋅min^−1^ versus ≈0.21, ≈0.31, and ≈0.99 bpm⋅min^−1^ for the moderate, heavy, and severe domains) [13].

As suggested by our predictive equation, these discrepancies may result from the different exercise intensities performed and/or ages. Indeed, in the current study, exercise intensities were anchored to the individual rest $\dot{V}$O_2_, gas exercise thresholds, and $\dot{V}$O_2max_, compared to previous studies in which different approaches were adopted, i.e., %GET, %ΔGET/$\dot{V}$O_2peak_ [14], or %$\dot{V}$O_2max_ [13]. These differences may have led to unmatched exercise intensities among studies and, in turn, to different _sc_HR amplitude.

To our knowledge, this is the first study directly comparing the _sc_HR between age-matched adults of both sexes across several exercise intensities from the moderate to severe domain. Our data show an _sc_HR reduced by aging (about 10% in our elderly versus young individuals) and one-third lower in females compared to males.

A possible explanation for the sex and age differences in the amplitude of the _sc_HR may partially derive from the metabolic heat production on the one hand and heat dissipation capacity on the other hand, both of which will affect the core temperature and, possibly, the _sc_HR over time [30,31,32]. The metabolic heat production for a given relative exercise intensity will be higher in individuals with a higher absolute $\dot{V}$O_2max_ (e.g., for trained vs. untrained, young vs. old, male vs. female, and heavier vs. lighter individuals) [30,32]. Indeed, males and females differ anthropometrically, with the first being generally heavier and taller, as well as displaying higher $\dot{V}$O_2max_ values [33]. In the present study, males were, on average, ≈17% heavier and ≈7% taller compared to female participants and, consequently, presented a ≈30% higher $\dot{V}$O_2max_ and ≈25% higher absolute $\dot{V}$O_2_ values at the fifth minute for each of the matched relative exercise intensity. Indeed, the $\dot{V}$O_2_ at the fifth minute was found to be a significant predictor of the _sc_HR (even if discarded from our predictive equation due to its cross-correlation with the relative exercise intensity to RCP), so much so that, when the individual _sc_HR were normalized for the $\dot{V}$O_2_ at the fifth minute, the sex difference disappeared (see Figure 2), suggesting an effect of the absolute oxygen uptake on the observed sex differences in the _sc_HR.

In addition, the ability to eliminate heat (dissipation) through vasodilation and sweating is affected by body dimensions (mainly related to the ratio between the body surface area and mass), sex, age, and aerobic fitness (that affect the threshold and sensitivity of the sweating mechanism) [30]. In particular, while the larger surface/mass ratio in women may represent an advantage for heat dissipation, the lower sweating rate vs. men may represent a disadvantage [30]. Unfortunately, no measures of body temperature, sweating rate, or peripheral blood flow were taken in our study; therefore, we have no means to quantify the overall effect of sex and body geometry on this variable. Notably, in a previous study, in which exercise intensities between sexes were matched for metabolic heat production, the anecdotal HR increments during exercise were similar in males vs. females (from the 30th to 90th minute of about 0.71 and 0.61 bpm⋅min^−1^, respectively) [30]. However, the authors reported HR values only above 30 min of exercise, when different factors (e.g., decreasing stroke volume and dehydration) started affecting HR dynamics in a well-known phenomenon called cardiovascular drift [34]. Thus, further studies need to examine the effect of heat production and dissipation on the HR slow component during the early phase of exercise (i.e., <15 min).

Similarly, the progressive decrease in the _sc_HR amplitude with aging can be partially due to the lower absolute $\dot{V}$O_2_ observed in older people compared to young individuals. A decrease in $\dot{V}$O_2max_ with aging is a well-documented phenomenon quantified by about ~10% every 10 years [18]. The aging $\dot{V}$O_2max_ decline may, therefore, be partially related to the lower _sc_HR that characterizes older individuals compared to young ones. In addition, reductions in the intrinsic maximal heart rate and in the chronotropic responsiveness to β-adrenergic stimulation with aging [35] reduce the HR reserve and might lower the potential for the HR excursion during exercise.

To summarize, we think it is fair to hypothesize that the metabolic heat production (i.e., absolute $\dot{V}$O_2_), the temperature regulation capacity, and the potential for a maximal HR excursion may play a role in the discrepancy observed between sexes and ages. However, further investigation is needed to fully understand the physiological underpinnings of the _sc_HR dynamics as well as to examine the HR dynamics over more prolonged exercise (i.e., >15 min of exercise) to better elucidate the interplay roles between the HR slow component and cardiovascular drift phenomenon.

Moreover, it remains to be determined if the menstrual cycle phase may affect the _sc_HR component’s dynamic, possibly explaining the differences between young and postmenopausal women. However, a previous study exploring the menstrual cycle’s effect on cardiovascular drift excludes such a possibility. In fact, the cycle phase affected the absolute HR at a given workload, yet not its time-dependent slow dynamic [36].

In conclusion, whenever exercise is anchored to HR targets, the presence of an _sc_HR that is partially dissociated from $\dot{V}$O_2_ will cause an undesired, time- and intensity-dependent reduction in work rate (approximately 6 and 14% over a moderate and heavy intensity 30 min session, respectively) during the training session [13,14,15]. Notably, at intensities closest to the exercise intensity boundaries (e.g., H1 in the present study), a time-dependent work rate decline could also cause an undesired switch to the lower-intensity exercise domain (i.e., from heavy-to-moderate or severe-to-heavy). Therefore, ignoring the _sc_HR when fixed HR targets are used for exercise prescription may be responsible for a reduction in the overall energy expenditure and a down-shift in the intensity domain with important implications for the health and training outcomes [5,13,14,15]. The practical impact of ignoring the HR slow component when prescribing exercise (anchored to a %HR_max_ or HR target) may be minimal or severe depending on its amplitude, which can differ across intensities and in different populations, fitness levels, or diseases [13,14,16,17]. Notably, previous studies showed that the uncertainty in exercise prescription deriving from the _sc_HR is not more severe in obese patients compared to healthy controls, as the _sc_HR amplitude was similar in each domain [15]. On the contrary, this issue seems to be less relevant in cardiac patients’ β-Blockers, as the reduction in workload over time during exercise at a fixed HR was less pronounced than in healthy individuals [17]. Further studies should be conducted on different patient populations.

Lastly, it should be noted that, whenever a measure of external load is possible, this can be used as a proxy of metabolic intensity to prescribe exercise. This is typically the case in cycling or running, where wearable sensors offer the opportunity to monitor the external load, although at a cost [5,6]. However, in a variety of situations (e.g., activities where the metabolic cost is affected by the terrain conditions, inclination, and skills of the individual), HR offers a reliable, time-resolved, and accessible index of metabolic intensity. Among the advantages of this approach is the immediate transferability of the metabolic intensity from a cardiorespiratory test to any health/rehabilitation center, gym, and external environment, and across different ergometers [10,11,12,37,38]. The decision to use either an external load (e.g., speed or power output) or a proxy of metabolic intensity (e.g., HR) ultimately depends upon the type and context of the activity, as well as the affordability of the instruments.

## 5. Practical Implications and Limitations

The adjustment of the HR target over time is made possible by our predictive equation with the following steps: (i) from a previous cardiopulmonary test, detect the desired metabolic intensity and the associated HR target; and, (ii) during the exercise training beyond the fifth minute, dynamically update the HR target based on the rate of increment (_sc_HR, i.e., bpm⋅min^−1^) estimated with our equation (i.e., based on the training intensity in %RCP, sex, and age of the individual). This procedure would grant that the desired stimulus is maintained throughout the exercise session in a given individual.

It should be noted that the developed predictive equation does not take into account factors that may potentially affect HR kinetics, such as fatigue, overtraining, nutrition, hydration, and environmental conditions such as temperature and humidity, which were controlled for in our study to the best of our abilities. In addition, the sample tested in the present study was representative of moderately active to active individuals. Thus, to confirm or refuse the absence of the fitness level’s effect on the _sc_HR’s dynamic, further studies are needed on a more heterogeneous population (e.g., sedentary individuals vs. elite athletes). Lastly, the assumption of a linear nature of the _sc_HR kinetics and, therefore, the validity of our predictive equation needs to be confirmed over longer exercise sessions.

## 6. Conclusions

The prediction equation for _sc_HR developed and validated in the current study provides the means to dynamically adapt HR targets over time, avoiding an undesired reduction in absolute and relative training load. This strategy would allow the maintenance of the desired metabolic stimulus throughout an exercise session in a heterogeneous population.

## Figures and Tables

**Figure 1 sports-13-00045-f001:**
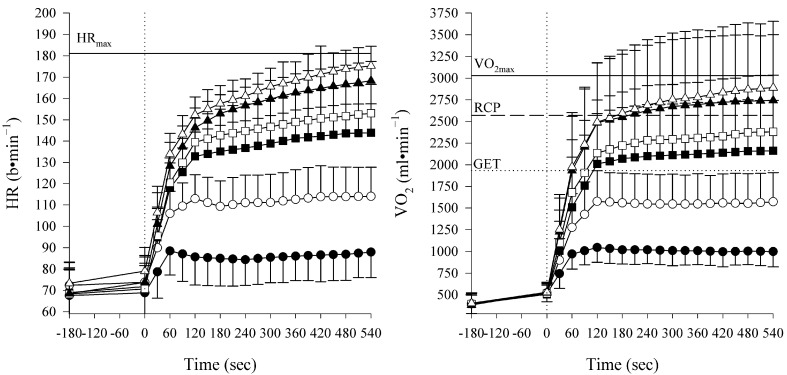
Data are expressed as mean ± SD of the heart rate (HR) and oxygen uptake ($\dot{V}$O_2_) during different constant-workload intensities (moderate: M1 ● and M2 ○; heavy: H1 ■ and H2 □; and severe trials: S1 ▲; S2 ∆). As identified from the ramp test, mean HR and $\dot{V}$O_2_ peak values (HR_max_ and $\dot{V}$O_2max_) are displayed as horizontal lines, respectively, in the left and right panels. Gas exchange threshold (GET) and respiratory compensation point (RCP) were plotted as, respectively, dotted line and dashed line.

**Figure 2 sports-13-00045-f002:**
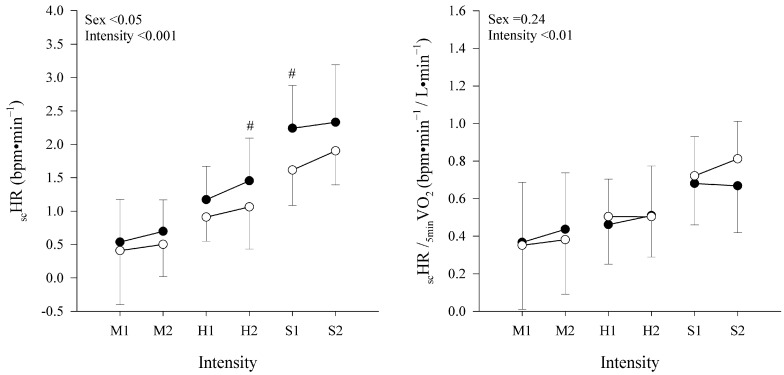
Absolute HR slow component (_sc_HR) and relative to the $\dot{V}$O_2_ at the 5th minute (_sc_HR/_5min_$\dot{V}$O_2_) are displayed as mean ± SD in, respectively, left and right panel. Data are shown at different constant-workload intensities (moderate: M1 and M2;heavy: H1 and H2;and severe trials: S1, and S2) in males (●) and females (○). Main effects of sex and intensity are displayed for both variables. # indicates a significant difference from females at the same exercise intensity.

**Figure 3 sports-13-00045-f003:**
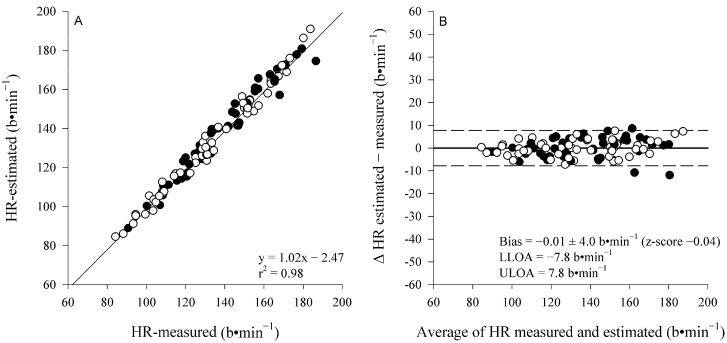
Pearson correlation and Bland–Altman analysis (panel **A** and **B**, respectively) between the measured and estimated HR at the end of each constant-work-rate exercise. Males (●) and females (○). Since no interaction effects (sex × method) were detected, cumulative coefficient of determination (r^2^), bias, z-score, and precision are shown.

**Table 1 sports-13-00045-t001:** Anagraphic, anthropometric, and cardiorespiratory variables of the Step 1 subjects.

Sex	n.	Age	BMI	HR_max_	%HR_max_	RPE	RER_max_	$\dot{V}$O_2max_	GET	RCP
		(yrs)	(kg⋅m^−2^)	(b⋅min^−1^)	%	(6–20 scale)		(mL⋅min^−1^⋅kg^−1^)	$\%\dot{V}$O_2max_	$\%\dot{V}$O_2max_
Males	31	33 ± 9	24 ± 2	182 ± 11	97 ± 2	19 ± 1	1.17 ± 0.05	51.0 ± 11.0	62.0 ± 6.8	86.2 ± 5.3
Females	34	37 ± 15	22 ± 2	180 ± 11	96 ± 2	19 ± 2	1.13 ± 0.04	41.4 ± 7.1	62.2 ± 7.1	83.2 ± 6.0
Tot.	65	35 ± 13	23 ± 2	181 ± 12	96 ± 2	19 ± 1	1.15 ± 0.05	45.5 ± 10.0	62.1 ± 7.1	84.5 ± 6.0

Data are expressed as mean ± SD: body mass index (BMI), maximal heart rate (HR_max_), percentage of age-predicted HR_max_ (%HR_max_), ratings of perceived exertion (RPE), maximal respiratory exchange ratio (RER_max_), relative maximal oxygen uptake to body weight ($\dot{V}$O_2max_), gas exchange threshold (GET), and respiratory compensation point (RCP) in males (M) and females (F).

**Table 2 sports-13-00045-t002:** Variables measured during the constant workload exercises.

Sex	CWR	PO	$\dot{V}$O_2_	HR	%RCP	RPE
		(Watt)	(L⋅min^−1^)	(b⋅min^−1^)	(%)	(6–20 Scale)
Males	M1	26.0 ± 16.1	1.06 ± 0.23	89.2 ± 9.0	33.3 ± 8.2	7.1 ± 1.0
	M2	96.2 ± 16.9	1.77 ± 0.36	111.5 ± 13.5	57.5 ± 12.3	8.0 ± 1.8
	H1	186.85 ± 35.9	2.58 ± 0.51	137.6 ± 10.4	80.4 ± 6.0	10.8 ± 2.5
	H2	212.4 ± 37.7	2.79 ± 0.56	143.5 ± 9.5	90.2 ± 6.2	13.1 ± 1.9
	S1	260.2 ± 44.4	3.27 ± 0.63	156.9 ± 12.3	105.4 ± 8.5	15.5 ± 1.8
	S2	268.4 ± 45.3	3.33 ± 0.64	161.8 ± 9.3	109.3 ± 10.5	17.5 ± 2.1
Females	M1	45.1 ± 29.9	0.99 ± 0.18	106.0 ± 15.2	49.5 ± 9.9	6.7 ± 0.8
	M2	73.0 ± 30.0	1.30 ± 0.17	118.8 ± 15.2	62.7 ± 10.0	7.9 ± 1.3
	H1	107.4 ± 42.5	1.67 ± 0.45	137.4 ± 21.2	80.4 ± 8.3	10.1 ± 1.6
	H2	127.8 ± 40.0	1.91 ± 0.47	150.0 ± 18.7	92.6 ± 7.0	11.4 ± 2.9
	S1	154.8 ± 40.5	2.12 ± 0.46	161.7 ± 13.1	105.3 ± 8.5	14.5 ± 1.6
	S2	165.5 ± 38.5	2.21 ± 0.47	167.1 ± 13.0	112.7 ± 10.6	14.9 ± 3.7
Intensity effect		<0.001	<0.001	<0.001	<0.001	<0.001
Sex effect		<0.01	<0.001	=0.11	=0.39	=0.28
Intensity X sex		=0.45	=0.31	=0.78	=0.81	=0.94

Data are expressed as mean ± SD: power output (PO), oxygen uptake ($\dot{V}$O_2_), heart rate (HR), relative intensity to RCP (%RCP), and ratings of perceived exertion (RPE) at the 5th minute during constant workload (CWR): M1, M2, H1, H2, S1, and S2 in males and females. The main effects for the relative exercise intensity and sex are shown in the bottom line of the table.

**Table 3 sports-13-00045-t003:** Anagraphic, anthropometric, and cardiorespiratory variables of the Step 2 subjects.

Sex	n.	Age	BMI	HR_max_	%HR_max_	$\dot{V}$O_2max_	GET	RCP
		(yrs)	(kg⋅m^−2^)	(b⋅min^−1^)	%	(mL⋅min^−1^·kg^−1^)	$\%\dot{V}$O_2max_	$\%\dot{V}$O_2max_
Males	18	47 ± 18	25 ± 2	170 ± 17	98 ± 1	40.6 ± 11.1	60.1 ± 7.2	84.2 ± 5.1
Females	18	47 ± 16	21 ± 2	175 ± 10	97 ± 3	40.8 ± 7.0	62.4 ± 9.9	82.2 ± 5.4
Tot.	36	47 ± 16	23 ± 3	173 ± 14	97 ± 2	40.7 ± 9.2	61.1 ± 8.5	83.1 ± 6.0

Data are expressed as means ± SD: age, height, weight, body mass index (BMI), peak heart rate (HR_peak_) relative maximal oxygen uptake to body weight ($\dot{V}$O_2max_), gas exchange threshold (GET), and respiratory compensation point (RCP).

## Data Availability

The data supporting the findings of this study are not publicly available due to privacy/ethical restrictions. The data are available from the corresponding author upon reasonable request.

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
