# Peer review of "Predicting Heart Rate Slow Component Dynamics: A Model Across Exercise Intensities, Age, and Sex†"

_sports, 2025, doi:10.3390/sports13020045_

Round 1

Reviewer 1 Report

Comments and Suggestions for Authors

Manuscript addresses an interesting and important topic about the heart rate slow component during constant-intensity exercise. Although the topic is relevant and has practical potential, the article requires significant additions and revisions to be considered a high-quality publication.

The introduction, while providing a good overview of heart rate as an indicator of exercise intensity, omits two key aspects. First, it lacks discussion about potential differences in scHR dynamics across various populations, such as older adults, children, or individuals with metabolic disorders. Second, it would benefit from emphasizing the practical implications of scHR for different types of physical activity - both in professional sports and in rehabilitation programs or public health initiatives. Additionally, the introduction should include clearly stated research questions posed by the authors before conducting the study. This would help readers better understand which aspects of scHR the authors aimed to investigate.

The results are interesting and show that scHR varies depending on exercise intensity, sex, and age, confirming the authors' hypotheses. However, the sample size is too small to consider these findings fully representative. This study should be described as preliminary or pilot research, which the authors should explicitly state.

The discussion, while correctly addressing the results, is very general and relies heavily on the same sources cited in the introduction. It lacks references to other studies that could provide alternative perspectives or broaden the context of the findings. In a scientific article, the discussion should be more detailed and multi-faceted, including potential methodological differences or contrasting results in other populations.

An additional issue is the presentation of study limitations and implications. Currently, this section is too brief and placed in an inappropriate location - it should be a separate paragraph preceding the conclusion. The authors should discuss the study's limitations in greater detail, such as the small sample size, lack of demographic diversity, and methodological constraints. They should also suggest specific directions for future research to fully explore scHR dynamics across different populations and conditions.

Unfortunately, IMO the article in its current form is relatively weak. It requires substantial rewriting. Only then could this study be considered a valuable contribution to knowledge in this field.

Author Response

Manuscript addresses an interesting and important topic about the heart rate slow component during constant-intensity exercise. Although the topic is relevant and has practical potential, the article requires significant additions and revisions to be considered a high-quality publication.

We thank the reviewer for the time and effort devoted to the revision of our work. We are glad that the reviewer finds merit in the manuscript, and we have done our best to address all the constructive criticism that was provided

The introduction, while providing a good overview of heart rate as an indicator of exercise intensity, omits two key aspects. First, it lacks discussion about potential differences in scHR dynamics across various populations, such as older adults, children, or individuals with metabolic disorders. Second, it would benefit from emphasizing the practical implications of scHR for different types of physical activity - both in professional sports and in rehabilitation programs or public health initiatives. Additionally, the introduction should include clearly stated research questions posed by the authors before conducting the study. This would help readers better understand which aspects of scHR the authors aimed to investigate.

We thank the reviewer for the suggestions. A deep revision has been made in many parts of the introduction, touching on all the reviewer's suggestions.

In particular, potential differences in scHR dynamics across various populations and the practical impact of ignoring the scHR have been better clarified (mainly from lines 77-108). As well we now state our initial hypothesis for the readers' clarity in lines 116-122 and report here for the reviewer's convenience:

“To this aim, we performed a two-step study: i) development of a comprehensive model for scHR prediction across different intensities, sexes, ages, and cardiorespiratory fitness and ii) testing the validity of the developed model using an independent sample of individuals. We hypothesized that intensity, sex, age, and cardiorespiratory fitness level would predictably affect the HR slow component, allowing the HR estimation over time in the independent sample. If confirmed, the developed mathematical model would allow for adjusting the prescribed HR targets over time by accounting for the individually estimated slow component of HR and, in turn, maintaining the desired metabolic stimulus during prolonged sessions across individuals.”

The results are interesting and show that scHR varies depending on exercise intensity, sex, and age, confirming the authors' hypotheses. However, the sample size is too small to consider these findings fully representative. This study should be described as preliminary or pilot research, which the authors should explicitly state.

We thank the reviewer for this comment. We could not agree more on the fact that more research is needed to provide a comprehensive solution to this underappreciated problem; future investigation needs to expand this research on different populations (e.g., adolescents) and in individuals with different diseases (e.g., Cardiac Patients β-Blockers).
Clearly, we acknowledge that our study has the limitations that have now been stated in the manuscript and confirmed by the reviewer. The question is, do these limitations void its usefulness?

Regarding the applicability of our findings in real life, the findings of our study could, in fact, find an immediate, direct application in the typical prescription scheme (accumulation of 30-60min per day of physical activity recommended to improve health and aerobic function, ACSM) for a broader population (at least in healthy individuals of both sexes of different ages). More research is needed to include different populations, diseases, and conditions (especially termal) however, we think that for standard exercise prescriptions that cover most of the populations and situations, the developed equation can result in a better predictive health outcome than a standard fixed HR approach.

We want to point out that one of the main focuses (and merits) of the current study is to test a broad spectrum of exercise intensity in different populations for the first time and to validate the developed model in an independent sample across domains. To this aim and from an ecological point of view, we picked the shortest possible trial duration that would allow us to identify a linear trend while at the same time containing the overall subject (that was based on power statistical analysis) and laboratory commitment to this project (i.e., 563 experimental trials for step 1 and 2). We think that the same amount of trials, with pre-test instruction, would be harder to test in some specific populations (e.g., children) and diseases. However, by confirming with our model that the scHR is linearly related to relative exercise intensity, the future analysis may now focus more on different aspects like exercise duration, condition, and in different populations. To this aim more detailed future directions and limitations have been proposed in the discussion section.

In conclusion, we think that the current article would contribute to the awareness of the prescription bias introduced by the HR slow component and provide a means to mitigate this problem, at least in short-duration exercises. While some articles have described “drift-induced” reduction of training dose over time in different populations (i.e., healthy males, obese, Cardiac Patients β-Blockers, and postmenopausal women (zuccarelli 2018, 2020, Baldassarre 2024, Teso 2022)) and in individuals with different levels of fitness (i.e., before and after bed rest,  Baldassarre 2022), this problem remains largely underappreciated outside of physiology laboratories.

We hope that the reviewer will agree that, within the limits that were clearly acknowledged and expanded now, this study does provide novel data and a practical tool to be used in at least some applications. Moreover, the current work can be expanded on by other scientists, contributing to overcome its limitations.

The discussion, while correctly addressing the results, is very general and relies heavily on the same sources cited in the introduction. It lacks references to other studies that could provide alternative perspectives or broaden the context of the findings. In a scientific article, the discussion should be more detailed and multi-faceted, including potential methodological differences or contrasting results in other populations.

We thank the reviewer for these suggestions. A deep revision of the discussion has been made in many parts, as suggested (mainly from line 383). We hope the added different future directions and the expanded potential mechanism underlying scHR are clearer and stated for the reader's acknowledgment.

An additional issue is the presentation of study limitations and implications. Currently, this section is too brief and placed in an inappropriate location - it should be a separate paragraph preceding the conclusion. The authors should discuss the study's limitations in greater detail, such as the small sample size, lack of demographic diversity, and methodological constraints. They should also suggest specific directions for future research to fully explore scHR dynamics across different populations and conditions.

We agree with the reviewer. Future directions have now been expanded and detailed on pages 11-12 within the discussion section. Moreover, an expanded limitations and implications section has been moved before the conclusion. Specific limitations have also been placed along the revised discussion.
Thanks for the suggestion!

Unfortunately, IMO the article in its current form is relatively weak. It requires substantial rewriting. Only then could this study be considered a valuable contribution to knowledge in this field.

We thank the reviewer for the careful revision and suggestions. We have done our best to address all the constructive criticism that was provided. We are confident that the quality of the article has been improved.
Thanks!

Reviewer 2 Report

Comments and Suggestions for Authors

This is a good study, and I have some questions to help improve this manuscript.

1. In Table 2, the interaction effects were not significant, why did the authors still conduct the tests for simple main effects or post hoc comparison, that is, within each sex testing between six intensities with post hoc comparison? If so, more literature or descriptions must be reviewed or elucidated to deduce the testing. Besides, lines 346-347, “Moreover, the study demonstrated for the first time a significant effect of sex and age on the amplitude of the scHR at a given intensity.” This sentence implied an interaction effect, so if the authors did not want to confuse readers, it is better to have more complete elucidation. In addition, lines 427-428, “…with its amplitude being larger with increasing intensity in males compared to females and in younger compared to older individuals.” It also implied the interaction effect. Therefore, in the Discussion section, it would be better to discuss more.

2. lines 93-94, “We hypothesized that age and sex would predictably affect the HR slow component in a given intensity domain.” This hypothesis was a little blurry, and it would be better to be elucidated.

3. lines 311-312, for equation 1, how were the males and females coded for the sex, such as males=1, female=0, or the opposite? It should be described in the content or denoted to help understand immediately.

4. line 313, what did SEE mean? It means “Standard error of estimates”? Denoting it would be better.

Author Response

This is a good study, and I have some questions to help improve this manuscript.

We thank the reviewer for the time and effort devoted to revising our work. We are glad that the reviewer finds merit in the manuscript, and we have done our best to address all the constructive criticism that was provided

    1) In Table 2, the interaction effects were not significant, why did the authors still conduct the tests for simple main effects or post hoc comparison, that is, within each sex testing between six intensities with post hoc comparison? If so, more literature or descriptions must be reviewed or elucidated to deduce the testing. Besides, lines 346-347, “Moreover, the study demonstrated for the first time a significant effect of sex and age on the amplitude of the scHR at a given intensity.” This sentence implied an interaction effect, so if the authors did not want to confuse readers, it is better to have more complete elucidation. In addition, lines 427-428, “…with its amplitude being larger with increasing intensity in males compared to females and in younger compared to older individuals.” It also implied the interaction effect. Therefore, in the Discussion section, it would be better to discuss more.

That's true; we agree with the reviewer. The presence of only main effects for sex and intensity (and no interaction) implies equal gaps in each (or most of) intensity between sex (and between intensities).
Indeed, with some exceptions (mainly in the M1 and M2 intensities), the post hoc did not show something different than the analysis for the main effect (e.g., absence in all intensity or presence in most of a sex effect, as detailed in Table 2).
Thus, we decided to delete the PostHoc analysis from Table 2 as it doesn’t add useful information and can confuse the readers, as the reviewer pointed out.

Moreover we delete the words “at a given intensity” from the sentences of lines 346-347 that imply an interaction effect.

In lines 427-428, we rephrase the sentence for the clearness of the reader:
“…with its amplitude being larger with increasing intensity and higher in males compared to females and in younger compared to older individuals.”

  1. lines 93-94, “We hypothesized that age and sex would predictably affect the HR slow component in a given intensity domain.” This hypothesis was a little blurry, and it would be better to be elucidated.

We thank the reviewer for this suggestion. A deep revision of the introduction has been made as requested also by other reviewers, and a more elucidated hypothesis has been described.

  1. lines 311-312, for equation 1, how were the males and females coded for the sex, such as males=1, female=0, or the opposite? It should be described in the content or denoted to help understand immediately.

Code has been detailed that on line 210 as males scored 0 and females scored 1.
However, for reader clarity, we also added this information in lines 311-312, as suggested by the reviewer.
Thank you!

  1. line 313, what did SEE mean? It means “Standard error of estimates”? Denoting it would be better.

Yes exactly! As suggested by the reviewer, we now denoted that by using the “full” form: “Standard error of estimates”
Thanks!

Reviewer 3 Report

Comments and Suggestions for Authors

General Comments:

This manuscript presents an insightful study on the heart rate slow component (scHR), addressing its variation across exercise intensities, age, and sex. While the content is scientifically significant, the presentation of data, methodology, and interpretation could be improved to enhance clarity, reproducibility, and impact.

Specific Comments:

Title and Abstract

Lines 1-31: The title is descriptive but could be more concise to highlight the predictive model and its application. Consider "Predicting Heart Rate Slow Component Dynamics: A Model Across Exercise Intensities, Age, and Sex."

Abstract: The methodology is explained adequately, but the potential applications of the model are underemphasized. Highlight practical applications for clinicians and trainers.

Introduction

Lines 37-89: While the introduction establishes the importance of scHR, it lacks a clear hypothesis statement. Adding a hypothesis at the end of the introduction would strengthen the transition to the methods.

Methods

Lines 95-133: The participant recruitment criteria are well described, but details on how participants were randomized into subgroups could improve transparency.

Lines 147-159: The rationale for selecting these specific intensities (% of V̇O2) is unclear. Explain why these thresholds are critical for modeling scHR.

Lines 171-179: Clarify how aberrant data points were identified and treated to address concerns about data preprocessing.

Results

Table 1 (Lines 238-241): The grouping of participants by age and sex is appropriate, but it would be beneficial to include a breakdown of fitness levels (e.g., trained vs. untrained).

Lines 303-312: The predictive equation is an important contribution but would benefit from a brief explanation of why specific variables were chosen or excluded during regression analysis.

Lines 323-333: The Bland-Altman analysis is explained well, but additional context on what constitutes "fair precision" would aid interpretation.

Discussion

Lines 341-409: The discussion links findings to existing literature effectively but does not explore potential mechanisms underlying the observed sex differences in scHR in sufficient depth. Expanding on physiological or hormonal factors could enrich this section.

Lines 407-409: The suggestion for further investigation is sound but could include more specific research directions (e.g., exploring environmental factors or longer durations).

Figures and Tables

Figures 1 and 2 (Lines 251-303): The figures are informative but lack clear legends. Ensure all abbreviations are defined directly in figure captions.

Practical Implications

Lines 425-433: While the implications are outlined, they remain somewhat theoretical. Include concrete examples of how practitioners might implement dynamic HR target adjustments using the predictive equation.

Limitations

Lines 434-437: The limitations section is well-crafted but omits the potential variability introduced by participant adherence to pre-test instructions (e.g., avoiding caffeine and exercise).

Language and Formatting:

Lines 12-31: Avoid overuse of technical jargon in the abstract to ensure accessibility for a broader audience.

References: Check consistency in reference formatting (e.g., citation style, use of DOI links).

Summary:

This study makes a significant contribution by developing a predictive model for scHR. However, improvements in clarity, methodological details, and discussion depth are needed. Addressing the outlined points will enhance the manuscript's quality and impact.

Author Response

General Comments:

This manuscript presents an insightful study on the heart rate slow component (scHR), addressing its variation across exercise intensities, age, and sex. While the content is scientifically significant, the presentation of data, methodology, and interpretation could be improved to enhance clarity, reproducibility, and impact.

 We thank the reviewer for the time and effort devoted to the revision of our work. We are glad that the reviewer finds merit in the manuscript, and we have done our best to address all the constructive criticism that was provided

Specific Comments:

Title and Abstract

 Lines 1-31: The title is descriptive but could be more concise to highlight the predictive model and its application. Consider "Predicting Heart Rate Slow Component Dynamics: A Model Across Exercise Intensities, Age, and Sex."

We accept and like the reviewer's suggestion. The title revision was made. Thanks!

Abstract: The methodology is explained adequately, but the potential applications of the model are underemphasized. Highlight practical applications for clinicians and trainers.

We thank the reviewer for this suggestion. We hope that the revised purpose and discussion in the abstracts will better describe the practical applications of this study.

Introduction

 Lines 37-89: While the introduction establishes the importance of scHR, it lacks a clear hypothesis statement. Adding a hypothesis at the end of the introduction would strengthen the transition to the methods.

We thank the reviewer for this suggestion. We state our initial hypothesis for the readers' clarity in lines 96-104 and report here for the reviewer's convenience:

“To this aim, we performed a two-step study: i) development of a comprehensive model for scHR prediction across different intensities, sexes, ages, and cardiorespiratory fitness and ii) testing the validity of the developed model using an independent sample of individuals. We hypothesized that intensity, sex, age, and cardiorespiratory fitness level would predictably affect the HR slow component, allowing the HR estimation over time in the independent sample. If confirmed, the developed mathematical model would allow for adjusting the prescribed HR targets over time by accounting for the individually estimated slow component of HR and, with that, maintain the desired metabolic stimulus during prolonged sessions in any individuals.”

Methods

 Lines 95-133: The participant recruitment criteria are well described, but details on how participants were randomized into subgroups could improve transparency.

Thanks for allowing us to clarify that. We rewrite the participants' section in lines 110-116. Reported below for the reviewer's convenience:

“The size required for each step was determined based on the power analysis reported above in the statistics analysis section. In addition, females and males were equally subdivided into three age groups (young <36, middle-aged between >36 and <55, and elderly >55 years (1)). As a result, sixty-five individuals (23 young, 12 females; 22 middle-aged, 12 females and 20 elderly, 10 females) participated in step 1, development of the prediction equation, while thirty-six (12 young, 6 females; 12 middle age, 6 females and 12 elderly 6 females) in step 2, validation of the prediction equation( (see respectively, Table 1 and Table 3 for participants' characteristics).

Lines 147-159: The rationale for selecting these specific intensities (% of V̇O2) is unclear. Explain why these thresholds are critical for modeling scHR.

Only two studies analyze the HR slow component across exercise domains (Zuccarelli 2018 and Teso 2022) (i.e.,  moderate, heavy, and severe) in young males and postmenopausal-only populations.
In both studies, the HR slow component was present even in the moderate domain, suggesting an intensity-dependent dynamic instead of a domain dynamic (contrary to the VO2 slow component, absent in the moderate, delayed steady state in the heavy and projection to VO2max in the severe).

Thus, to evaluate the effect of the intensity (within and between domains) and have a full characterization of the HR slow component, we performed two intensities per domain (one in the lower: 1/3, 33%, and one in the upper range: 2/3 66% of each domain). 

Lastly, the specific intensities (i.e., 33 and 66%) were chosen to be of a relevant distance from each other and from the boundaries of the domains (i.e., threshold: GET and RCP ). Having multiple points (the six intensities) spaced evenly across the entire spectrum of intensities that can be prescribed for aerobic exercise helps the development and accuracy of the linear regression model.

Lines 171-179: Clarify how aberrant data points were identified and treated to address concerns about data preprocessing.

We thank the reviewer for the careful revision. Additional informations were added to the manuscript in lines 186-193 and reported here for the reviewer's convenience:

"aberrant data points (that lay 3 standard deviations away from the local mean) were removed (24). This was accomplished using a linear least-squares regression method whereby the baseline (fitting window, approximately −180 to 0 s) and steady-state period (fitting window, approximately 180end trial) were fitted. A 99% prediction bands were used to identify any data points that lay three SD from the local mean. Care was taken not to delete data in the early portion of the transition (i.e., <180s);."

Results

Table 1 (Lines 238-241): The grouping of participants by age and sex is appropriate, but it would be beneficial to include a breakdown of fitness levels (e.g., trained vs. untrained).

We thank the reviewer for this suggestion. However, how detailed in lines 249 and 358-359. The subjects enrolled in the study were mostly active individuals (even if there was no inclusion criterion on this). Thus, we prefer not to include a breakdown of fitness level as a distinct subgroup of non-fit is absent.
Actually, we consider that a possible limitation of this study (that had the initial aim to test the fitness level effect on the scHR) therefore, we enlighten that on the discussion lines 465-468, and reported here for the reviewer’s convenience:

"In addition, the sample tested in the present study was representative of moderately-active to active individuals. Thus, to confirm or refuse the absence of the fitness level’s effect on the scHR’s dynamic, further studies are needed on a more heterogeneous population (e.g., sedentary individuals vs. elite athletes)."

Lines 303-312: The predictive equation is an important contribution but would benefit from a brief explanation of why specific variables were chosen or excluded during regression analysis.

We now add this information to the Statistical Analysis sections 228-230. And on the results, sections 309-312.

Briefly explanation for the reviewer:

After identifying the potential predictors of the scHR, cross-correlation analyses were tested. We detected that:
Age and HRmax were cross-correlated (r2>0.80 and variance inflation factor >5)
Age, %RCP, and V̇O2max to body weight were cross-correlated (r2>0.70 and variance inflation factor >10)
%RCP and V̇O2 at the 5th min were cross-correlated (r2>0.70 and variance inflation factor >10)

Between them, we choose %RCP and age as they lead to the highest R-squared and p-values in the final equation

Lines 323-333: The Bland-Altman analysis is explained well, but additional context on what constitutes "fair precision" would aid interpretation.

 The day-to-day variability in heart rate (HR) during steady-state exercise was found in other studies, being an average of 3.3%. In the present study the measured day-to day variability was slightly less (2.7%). This means that if the mean HR is 140 bpm (averaged of our intensities), it will typically fall within 5 bpm of the mean at a given work rate 95% of the time. Our Bias (mean +-SD: 0.1 +- 4.1 bpm*min-1) for the estimated vs measured HR in figure 3 was below the intrinsic day-to-day variability of the measure. For this reason we think it is “fair” to say “fair” imprecision. We added this information in lines 353

Discussion

 Lines 341-409: The discussion links findings to existing literature effectively but does not explore potential mechanisms underlying the observed sex differences in scHR in sufficient depth. Expanding on physiological or hormonal factors could enrich this section.
Lines 407-409: The suggestion for further investigation is sound but could include more specific research directions (e.g., exploring environmental factors or longer durations).

We thank the reviewer for these suggestions. A deep revision of the discussion has been made, as reviewers suggested. We hope that the added different future directions and the potential mechanism underlying scHR are now clearer and stated for the reader's acknowledgment.

Figures and Tables

 Figures 1 and 2 (Lines 251-303): The figures are informative but lack clear legends. Ensure all abbreviations are defined directly in figure captions.

Corrected! Thanks!

Practical Implications

 Lines 425-433: While the implications are outlined, they remain somewhat theoretical. Include concrete examples of how practitioners might implement dynamic HR target adjustments using the predictive equation.

We thank the reviewer for this suggestion. We now expand this information from line 453 to 460.

Limitations

 Lines 434-437: The limitations section is well-crafted but omits the potential variability introduced by participant adherence to pre-test instructions (e.g., avoiding caffeine and exercise).

The limitations section has now been expanded to implement all the reviewers' suggestion and reported below for convenience:

"However, the developed predictive equation does not take into account factors that may potentially affect HR kinetics, such as fatigue, overtraining, nutrition, hydration, and environmental conditions such as temperature and humidity, which were controlled for in our study to the best of our abilities. In addition, the sample tested in the present study was representative of moderately active to active individuals. Thus, to confirm or refuse the absence of the fitness level’s effect on the scHR’s dynamic, further studies are needed on a more heterogeneous population (e.g., sedentary individuals vs. elite athletes). Lastly, the assumption of a linear nature of the scHR kinetics and, therefore, the validity of our predictive equation needs to be confirmed over longer exercise sessions."

Language and Formatting:

Lines 12-31: Avoid overuse of technical jargon in the abstract to ensure accessibility for a broader audience.

The abstract has now been deeply revised. We hope the reviewer would find that more clear for a broader audience.

References: Check consistency in reference formatting (e.g., citation style, use of DOI links).
Citation style and DOI links have now been revised. Thanks!

Summary:

This study makes a significant contribution by developing a predictive model for scHR. However, improvements in clarity, methodological details, and discussion depth are needed. Addressing the outlined points will enhance the manuscript's quality and impact.

We thank the reviewer again for the careful revision and suggestions.

Round 2

Reviewer 1 Report

Comments and Suggestions for Authors

Thank you for your comment on the review.

Reviewer 2 Report

Comments and Suggestions for Authors

OK

Reviewer 3 Report

Comments and Suggestions for Authors

All concerns have been addressed.